# Development of a Novel Piezoelectric Actuator Based on Stick–Slip Principle by Using Asymmetric Constraint

**DOI:** 10.3390/mi14061140

**Published:** 2023-05-28

**Authors:** Liang Wang, Heran Wang, Junxiang Jiang, Tianwen Luo

**Affiliations:** 1School of Mechanical and Civil Engineering, Jilin Agricultural Science and Technology University, Jilin 132101, China; luotw314@sina.com; 2School of Mechanical Engineering, Northeast Electric Power University, Jilin 132012, China; wliang0730@163.com (L.W.); 2202101039@neepu.edu.cn (H.W.)

**Keywords:** piezoelectric stick–slip, asymmetric constraints, flexible hinge

## Abstract

In this work, a novel piezoelectric actuator based on the stick–slip principle is proposed. The actuator is constrained by an asymmetric constraint approach; the driving foot produces lateral and longitudinal coupling displacements when the piezo stack is extended. The lateral displacement is used to drive the slider and the longitudinal displacement is used to compress the slider. The stator part of the proposed actuator is illustrated and designed by simulation. The operating principle of the proposed actuator is described in detail. The feasibility of the proposed actuator is verified by theoretical analysis and finite element simulation. A prototype is fabricated and some experiments are carried out to study the proposed actuator’s performance. The experimental results show that the maximum output speed of the actuator is 3680 μm/s when the locking force is 1 N under the voltage of 100 V and frequency of 780 Hz. The maximum output force is 3.1 N when the locking force is 3 N. The displacement resolution of the prototype is measured as 60 nm under the voltage of 15.8 V, frequency of 780 Hz and locking force of 1 N.

## 1. Introduction

With the rapid development in precise positioning, it is difficult for power-transmitting actuators to meet the nanopositioning requirement. Actuators with nanometer resolution occupy an important position in modern society [1,2]. A piezoelectric actuator is a micro-actuator that converts electrical energy into mechanical energy by the inverse piezoelectric effect of piezoelectric material. It plays an important role in many fields [3,4,5], such as precision machining [6], optical focusing [7], scanning probe microscopy [8] and robotics [9], because of its high accuracy [10], lack of electromagnetic interference [11], small size [12,13] and high response speed [14,15]. A piezoelectric actuator can be divided into a resonant type and non-resonant type from the operating condition [16,17,18]. The resonant type, also known as an ultrasonic actuator, can achieve high speed movement [19]. However, it has serious wear and heat problems that lead to a lower accuracy compared to the non-resonant type actuator [20]. A non-resonant actuator can be divided into a direct drive type, inchworm type and inertial drive type [21,22,23].

A piezoelectric stick–slip actuator is the main type of the inertial drive type, which consists of a drive unit and a moving unit [24]. The principle is to drive the slider movement by using the transformation between the static and dynamic friction forces [25,26]. They are widely studied due to their simple control, smooth output, long stroke and nanometer resolution [27,28,29]. The principle of the piezoelectric stick–slip actuator with a flexure hinge is that lateral and longitudinal displacements are produced at the driving foot when the piezo stack is elongated [30]. The lateral displacement drives the slider by the friction force, while the longitudinal displacement compresses the slider and increases the friction force between the driving foot and the slider [31]. Based on the above working principle, scholars have proposed a variety of structural asymmetric flexible hinge piezoelectric stick–slip actuators. For example, a lever-type asymmetric piezoelectric stick–slip actuator was proposed by Yang et al. [32]. The experimental results show that the locking force reaches 2.16 N and the maximum output speed is 283 μm/s. Li et al. proposed a piezoelectric actuator that used a combination of a lever-type flexible hinge and a triangular flexible hinge [33]. The proposed actuator can amplify the clamping and driving forces through the triangular flexible hinge. The drive speed performance is also improved by increasing the design angle. Lu et al. designed a piezoelectric stick–slip actuator, which combined an asymmetric bending hinge and a triangular displacement amplification mechanism [34]. The proposed actuator can achieve high speeds at lower frequencies, and the maximum stepping efficiency of the prototype is 97.9% with the maximum load capacity of 2.4 N and the maximum output speed of 20.17 mm/s.

The key to a piezoelectric stick–slip actuator using a flexure hinge is to generate horizontal displacement and vertical displacement at the same time. An asymmetric structure is always used to design the flexure stator. Such an asymmetric structure frequently results in the increase in the whole size, which is not beneficial to further miniaturization. For example, for the rectangle flexure stator, two sides should be set obliquely as a parallelogram to generate horizontal displacement and vertical displacement simultaneously. Such a parallelogram will increase the whole size along the horizontal direction. In this work, a piezoelectric stick–slip actuator with an asymmetric constraint is proposed, which is different from an asymmetric structure. For the rectangle flexure stator, this proposal generates horizontal displacement and vertical displacement by unchanging the whole size of the stator, which is quietly beneficial for the miniaturization of the piezoelectric actuator. The configuration and operating principle of the proposed actuator are discussed in Section 2. In Section 3, theoretical modeling and simulation analysis are performed. In Section 4, the experimental test system of the actuator and a series of experimental results are presented. Finally, the conclusions are presented in Section 5.

## 2. Structure and Principle

The proposed actuator is asymmetric and its deformation capability is provided by two flexible hinges. Therefore, the structural design of the proposed actuator is critical. The height *h* of the right-side step needs to be determined by simulation to achieve the maximum displacement generated by the driving foot. Therefore, this section details the structural design of the proposed actuator and the simulation to determine the value of *h* of the actuator. In addition, the operating principle of the actuator is described in detail.

Figure 1a shows the three-dimensional structure of the designed piezoelectric stick–slip actuator. It mainly consists of a slider, a stator and a moving platform. It can be clearly seen that the position of the fixation hole on the right side of the stator varies with *h*. Therefore, the two fixed holes are constrained in an asymmetric method. The stator is fixed to the moving platform by two preload bolts, and the moving platform is moved axially by adjusting the micrometer knob, which causes the driving foot to contact and preload the slider. The detailed structure of the stator is shown in Figure 1b. The driving foot is located in the middle of the top of the stator and contacts with the slider. The piezo stack is located directly below the driving foot and it is inserted into the flexible stator by a shim and a bolt. The assembly details of the stator and the moving platform are shown in Figure 1c. A platform spacer is added between the stator and the moving platform. A thickened plate is added between the slider and the base to allow the driving foot to contact the slider.

In order to improve the output performance, the statics simulation of the stator is performed to determine the height *h* of the right side shown in Figure 1b. During the simulation, a displacement of 10 μm is applied along the Y axis below the driving foot, which imitates the deformation of the piezo stack. Figure 2 shows the simulation results of the driving foot displacement corresponding to different heights *h*. The X-direction displacement and Y-direction displacement data of the driving foot are obtained for different asymmetric stator structures with *h* from 0 to 13 mm. It can be seen from Figure 2 that the range of variation of the displacement in the Y direction is smaller as *h* increases. The X-direction displacement appears as a peak as *h* increases; furthermore, the X direction is the driving direction. In other words, the actuator will output good performance if the driving foot generates a large drive distance in the X direction. It can be seen from Figure 2 that the maximum lateral displacement of the driving foot is 8.9 μm as *h* is 9 mm. Therefore, *h* is determined to be 9 mm in this work.

The motion process of the actuator will be studied. Figure 3 shows the operating principle of the proposed piezoelectric actuator. Figure 3a shows a sawtooth waveform excitation signal. When time t = *t*_0_, the slider and the driving foot contact due to the initial preload force. The longitudinal output displacement of the piezo stack generates the lateral and longitudinal coupling displacement on the driving foot when a sawtooth waveform signal is applied to the piezo stack. The lateral displacement drives the slider movement and the longitudinal displacement increases the pressure between the slider and the driving foot. Each motion cycle is divided into two steps.

Step 1: From time *t*_0_ to *t*_1_, the piezo stack is slowly stretched longitudinally. The deformation of the piezo stack causes the driving foot to obtain two displacements in the X and Y axes. The displacement of the driving foot in the X axis direction is used to push the slider in the positive direction of the X axis, and the static friction is the driving force for the slider to move forward. The displacement of the driving foot in the Y axis direction is used to increase the pressure between the slider and driving foot. The static friction force generated in this stage pushes the slider forward by a distance *d*_1_, as shown in Figure 3b.

Step 2: From time *t*_1_ to *t*_2_, the piezo stack rapidly recovers its initial length. The driving foot returns to its initial position due to the elastic recovery of the flexible hinge. The slider will move to the position shown in Figure 3c. As the pressure between the driving foot and the slider rapidly decreases, the resulting dynamic friction will push the slider to move in the negative direction of the X axis. In this stage, the dynamic friction force is the resistance.

After the above process, the slider is moved by a displacement of *d*_2_ in the positive direction along the X axis relative to the initial position. By repeating steps 1 and 2, the proposed actuator gradually performs a large stroke motion along the positive direction of the X axis.

## 3. Simulations and Analyses

To verify the feasibility of the proposed piezoelectric actuator, theoretical model and finite element methods are used. The most critical factor affecting the output performance of the actuator is the lateral displacement on the driving foot. Therefore, a geometric relationship model is built to account for the change in motion of the driving foot. The variation of the displacement at the top point of the driving foot is studied, as shown in Figure 4. Point *A* in the figure represents the driving foot vertex, and *BC* represents the length of the flexible hinge *h* on the right side of the asymmetric stator. In step 1, when the piezo stack is slowly elongated, the right side of the stator rotates counterclockwise around point *C*. Point *A* will move to the position of *A’* and point *B* will move to the position of *B’*.

The following equation can be derived from the geometric relationship in Figure 4.
(1)$$A^{\prime }C=\sqrt{A^{\prime }{B^{\prime }}^{2}+B^{\prime }C^{2}}.$$
(2)$$A^{\prime }E=\sqrt{A^{\prime }C^{2}-EC^{2}}.$$

Its own deformation can be neglected due to the stiffness of the flexible hinge.
(3)$$A^{\prime }B^{\prime }=AB,B^{\prime }C=BC.$$

Some geometric equations can be listed:(4)AB=DE,
(5)BE=AD,
(6)EC=BE+BC=AD+BC.

From the above equation, the lateral displacement *AD*′ can be deduced.
(7)$$A^{\prime }D=AB-A^{\prime }E=AB-\sqrt{\sqrt{{\left(AB^{2}+BC^{2}\right)}^{2}}-{\left(AD+BC\right)}^{2}}.$$

In the above equation, the values of *AB*, *BC* and *AD* are 11 mm, 10.3 mm and 10 μm, respectively. The purpose of this geometric model is to obtain the lateral displacement *A′D* on the driving foot by a geometric solution. The result of *A′D* is 9.4 μm when the longitudinal displacement *AD* is 10 μm. This theoretical approach is combined with the next simulation to verify the design feasibility of this actuator.

Statics analysis is performed by ANSYS software. The material of the flexible mechanism is AL7075 with the following properties: density of 2810 kg/m^3^, modulus of elasticity of 71.7 GPa and Poisson’s ratio of 0.33. Element Solid187 is used. To analyze the deformation of the asymmetric actuator under the applied sawtooth wave signal, a finite element analysis of the stator is performed. Fixed constraints are applied to the two holes on the stator, and a 10 μm displacement along the positive Y axis is applied at the location of the upper surface of the piezo stack. As shown in Figure 5, the displacements of the driving foot along the X and Y directions are *d*_x_ = 8.9 μm and *d*_y_ = 9.9 μm, respectively.

In addition, the stress analysis of the actuator is carried out by ANSYS software. The analysis result is illustrated in Figure 6. The analysis result shows that the obtained maximum stresses are 40.6 MPa, 38.9 MPa and 16.1 MPa along the X, Y and Z directions under the applied displacement of 10 μm, respectively. The maximum von Misesstress is 34.8 MPa and these values are far smaller than the yield strength of material AL7075, which validates the reliability of the proposed actuator.

## 4. Experiment and Discussion

The proposed stick–slip piezoelectric actuator is fabricated and the prototype is shown in Figure 7. The size of the piezo stack (PSt150/5×5/20 L) is 6 × 6 × 18 mm^3^. The flexible stator is machined by wire-cutting technology and the cross-roller guideway is served as the slider. In the experiment, the load is tested by attaching some mass to the slider using a string through a pulley.

The prototype of the proposed actuator is fabricated and its mechanical output characteristics are tested according to the experimental system shown in Figure 8. The experimental system is described in detail below. The excitation signal of the sawtooth wave is generated by a waveform generator (WF1974, NF Corp., Japan) and then amplified by a power amplifier (HSA4051, NF Corp., Japan). The main parameters of the used power amplifier are the frequency range is DC~500 kHz, output voltage is 300 V_p-p_, output current is 2.83 A_p-p_ and conversion rate is 450 V/μs. The amplified signal is applied to the prototype and the laser displacement sensor (LK-H020, KEYENCE Corp., Japan) is used to measure the velocity and the displacement of the slider. The resolution of the used sensor is 5 nm and the measurement repeated accuracy is 20 nm. Finally, the experimental results are viewed and processed on a computer. A series of performance experiments are conducted to verify the feasibility of the proposed piezoelectric actuator.

The performance of the developed actuator is measured to obtain the performance between the output speed and excitation frequency under different locking forces. The experimental results are plotted in Figure 9. The excitation voltage is 100 V and the duty ratio is set as 90%. The locking force between the driving foot and the slider is 1 N, 2 N and 3 N. From the results, the following conclusions can be drawn: the output speed increases first and then decreases when the frequency increases, and the optimal frequency is 780 Hz. This can be explained as the relationship between the speed *v* and the frequency *f* which can be expressed as *v* = *f*·*d*. Here, *d* is the stepping distance of the slider. In general, the speed *v* is proportional to the frequency *f* if the stepping distance *d* remains unchanged. However, the exciting time of the sawtooth signal will decrease obviously when the frequency *f* increases. It means that the PZT stack will be shortened before it is fully extended, which results in the stepping distance *d* decreasing accordingly. This phenomenon will become more prominent if the frequency *f* continues to increase. On the whole, the frequency *f* increases and the stepping distance *d* decreases at the same time. The speed (*v* = *f*·*d*) will reach the maximum value when the frequency increases to a certain value. The performance of the actuator is better at a frequency of 780 Hz, so the optimal excitation frequency is 780 Hz. The actuator reaches the maximum output speed of 3680 μm/s when the locking force is 1 N, and the output speeds are 3580 μm/s and 3370 μm/s when the locking forces are 2 N and 3 N, respectively.

The frequency is set as 780 Hz, the voltage is 100 V and the duty cycle is 90%. The slider motion is tested under different locking forces, as shown in Figure 10. These experimental results show that the developed actuator outputs a stepping motion, which is consistent with the working principle. This validates the working principle and feasibility of the proposed piezoelectric stick–slip actuator with an asymmetric constraint. In addition, there is a negative displacement motion of the slider at the locking force of 2 N and 3 N. This is caused by the dynamic friction between the drive foot and the slider, which is consistent with the working principle of step 2. It can be seen that there is no displacement backward movement of the slider when the locking force is 1 N. This also shows that the actuator will have the maximum output speed at the locking force of 1 N.

Then, the relationship between the voltage and the output speed is tested at different locking forces. As shown in Figure 11, the slider starts to be driven when the voltage reaches 10 V, 10 V and 30 V at the locking force of 1 N, 2 N and 3 N, respectively. The experimental results show that the output speed increases with the input voltage, and both reach the maximum speed at a voltage of 100 V. The used piezo stack is recommended to work under the maximum voltage of 120 V. Therefore, the maximum voltage used in the experiment is 100 V in this work. The curves appear in the same form for different locking forces. Therefore, the voltage is adjusted to 100 V in the later experiments in order to make the actuator’s output performance better.

The load capacity of the proposed actuator is tested under a drive frequency of 780 Hz and a voltage of 100 V for different locking forces. As shown in Figure 12, the output speed decreases as the load mass increases. The output speed is 0 when the load reaches a certain value, which is the maximum output force under the current conditions. The experimental results show that the maximum output force of the actuator is 1.5 N when the locking force is 1 N, and the output force is 2.1 N and 3.1 N when the locking force is 2 N and 3 N, respectively.

The displacement resolution performance of the developed actuator was measured with 20 steps. The tested result is illustrated in Figure 13. The tested result shows clearly that the slider is actuated with the minimum step size of about 60 nm and it exhibits good repeatability by 20 steps. The developed actuator obtained the displacement resolution of 60 nm under the voltage of 15.8 V, frequency of 780 Hz and locking force of 1 N.

## 5. Conclusions

In this work, a novel piezoelectric stick–slip actuator based on asymmetric constraints is proposed and designed. The operating principle of the actuator is explained. The deformation and output displacement are studied by theoretical and simulation analyses to find the best form of flexible stator. In order to test the output performance of the slider, a prototype of the proposed piezoelectric actuator is fabricated and the experimental test system is built. The experimental results show that the maximum output speed of the prototype is 3680 μm/s when the voltage and driving frequency are 100 V and 780 Hz. The resolution of the prototype reached 60 nm under the locking force of 1 N. The maximum output force of the prototype was 3.1 N when the locking force was 3 N. These results show the feasibility of the proposal for designing a piezoelectric stick–slip actuator with a flexure hinge by using an asymmetric constraint. This design idea is different from that using an asymmetric structure and it has the merit of a compact size, which is quietly beneficial for further miniaturization.

## Figures and Tables

**Figure 1 micromachines-14-01140-f001:**
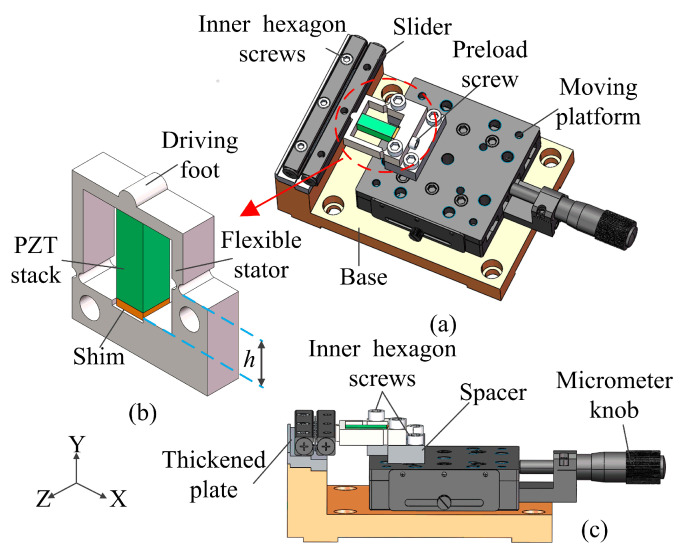
The configuration of the proposed piezoelectric stick–slip actuator. (**a**) Three-dimensional view of the actuator. (**b**) Stator. (**c**) Assembly relationship diagram of the actuator.

**Figure 2 micromachines-14-01140-f002:**
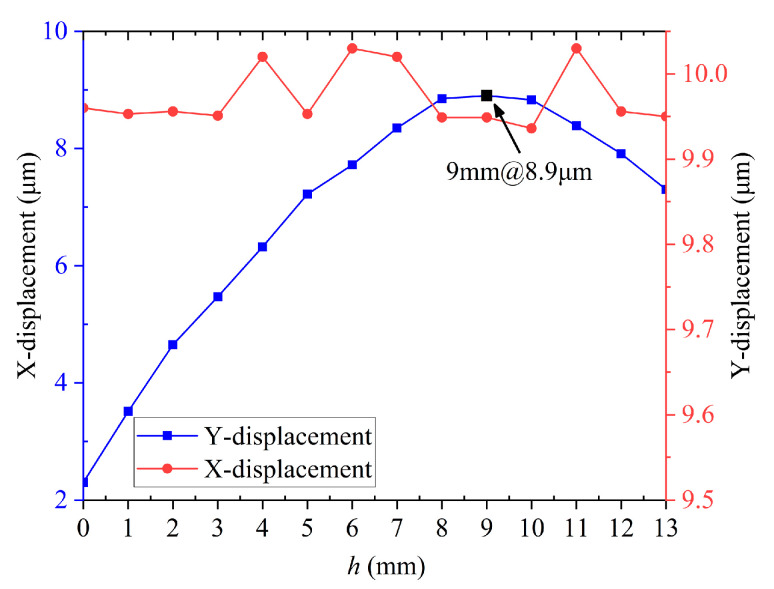
The relationship between the lateral displacement of the driving foot and *h*.

**Figure 3 micromachines-14-01140-f003:**
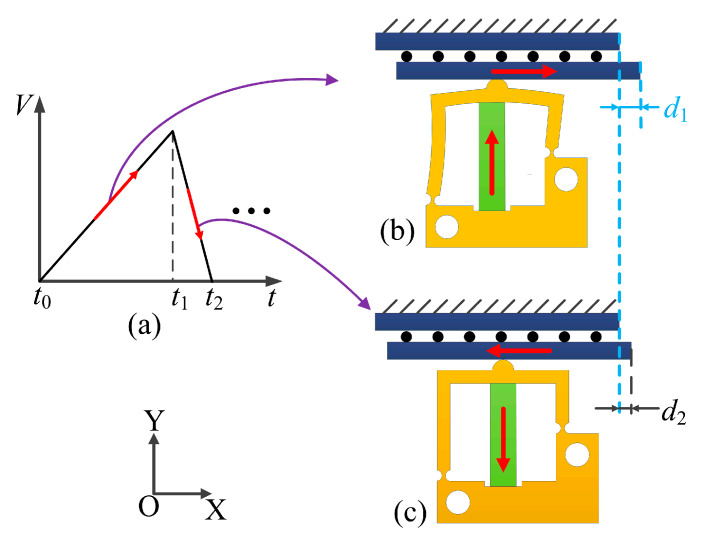
Operating principle of the proposed stick–slip actuator. (**a**) Applied sawtooth exciting signal. (**b**) Principle in step 1. (**c**) Principle in step 2.

**Figure 4 micromachines-14-01140-f004:**
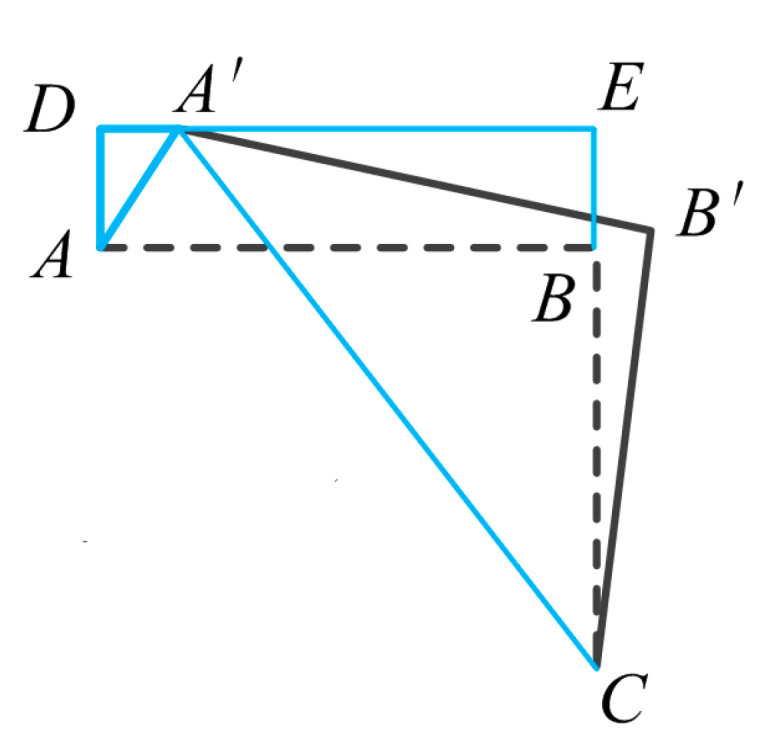
Schematic diagram of the motion geometry model.

**Figure 5 micromachines-14-01140-f005:**
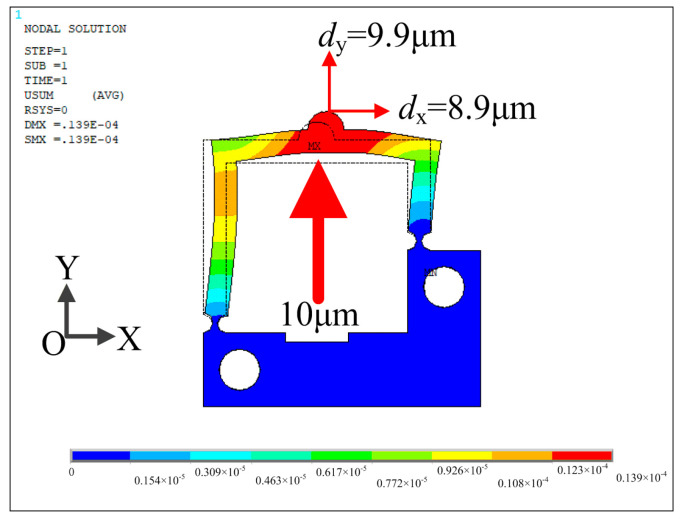
Finite element analysis result of the actuator in the working direction.

**Figure 6 micromachines-14-01140-f006:**
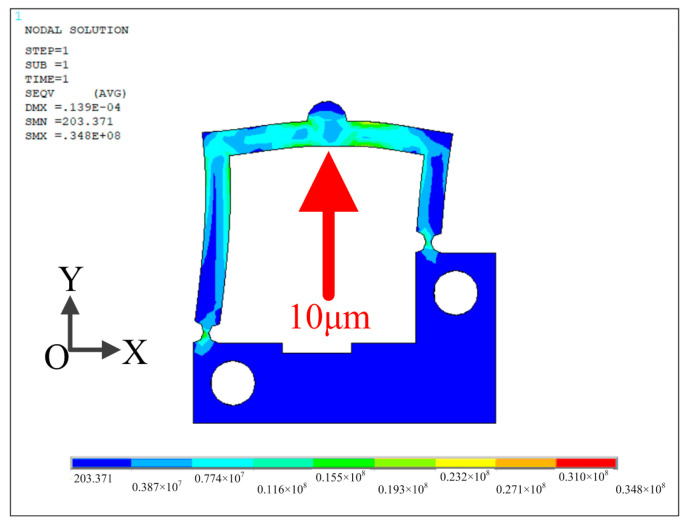
Stress analysis result of the actuator.

**Figure 7 micromachines-14-01140-f007:**
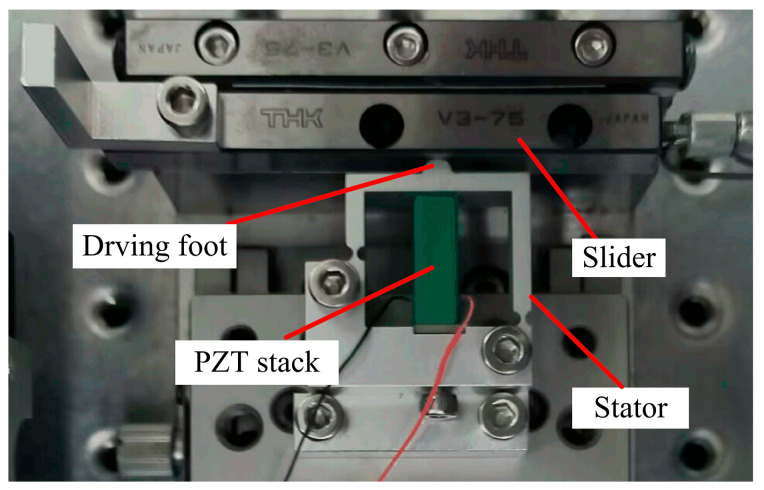
Prototype of the proposed piezoelectric actuator.

**Figure 8 micromachines-14-01140-f008:**
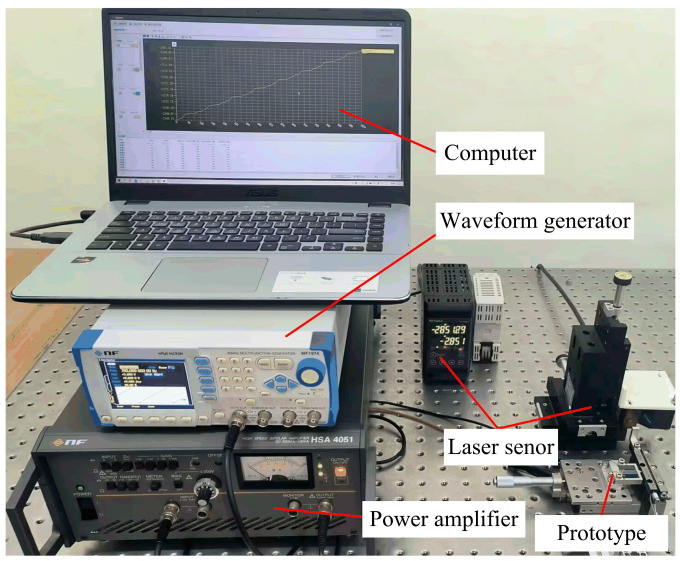
Experimental test system for the proposed actuator.

**Figure 9 micromachines-14-01140-f009:**
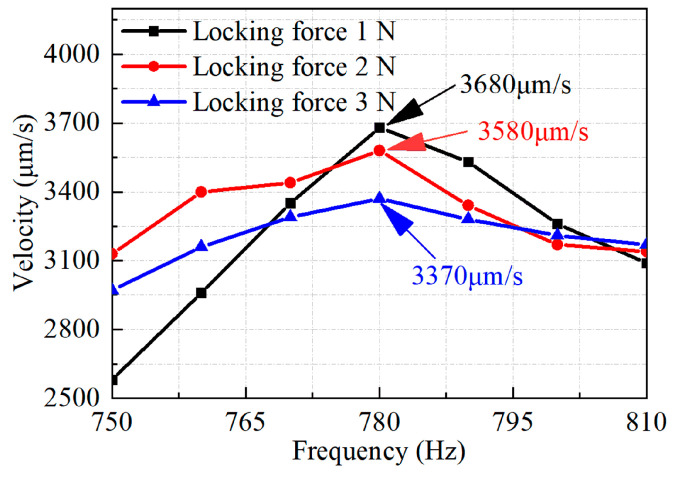
Tested results of the output velocity versus drive frequency.

**Figure 10 micromachines-14-01140-f010:**
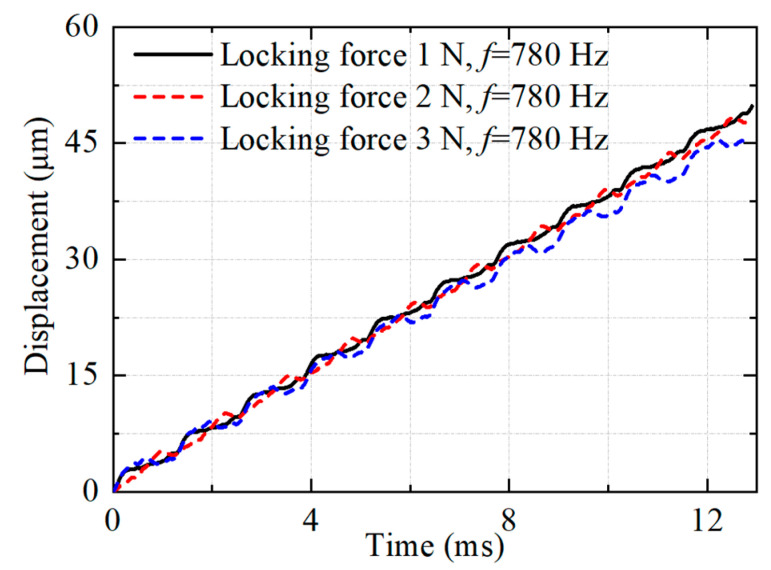
Tested results of the slider motion.

**Figure 11 micromachines-14-01140-f011:**
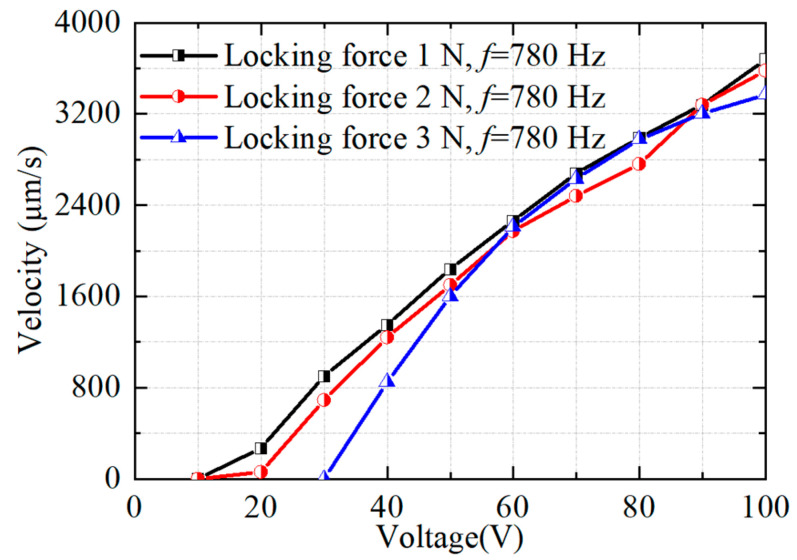
Tested results of the output velocity versus voltage.

**Figure 12 micromachines-14-01140-f012:**
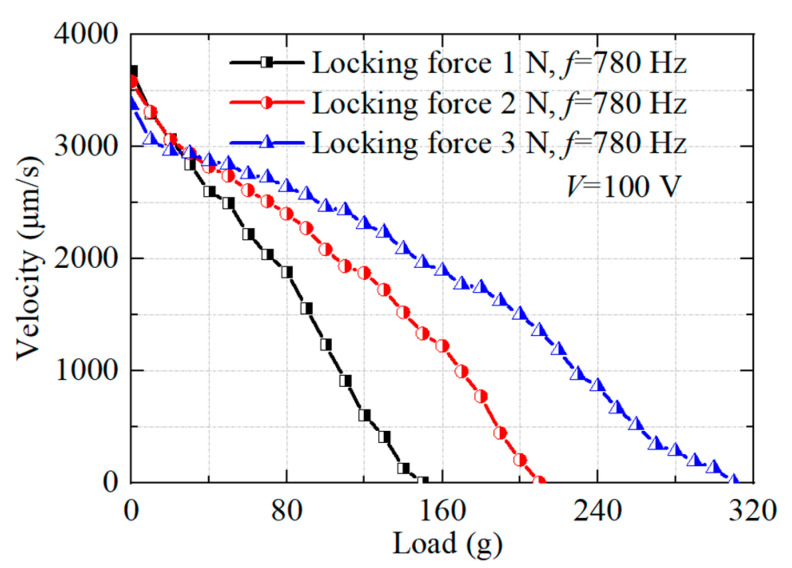
Tested results of force-loading capacity.

**Figure 13 micromachines-14-01140-f013:**
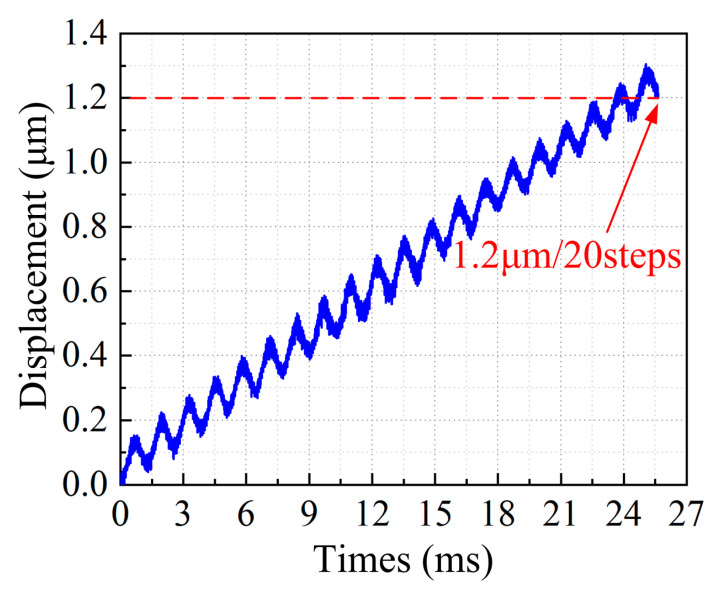
Resolution result of the developed actuator.

## Data Availability

The research data can be obtained for reasonable request by contacting with the corresponding author.

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
