# Peer review of "Development of a Novel Piezoelectric Actuator Based on Stick–Slip Principle by Using Asymmetric Constraint"

_micromachines, 2023, doi:10.3390/mi14061140_

Round 1
Reviewer 1 Report
I think that this article covers very important of micro/nano positioning systems, but I have a few comments on that:
1. Did you consider the magnitostrictive devices for positioning applications and what is the advantages/drawbacks of presented mechanisms in comparison with magnetostriction type of actuation? For best my knowledge they could be applied for subnanometer positioning.
2. Did you perform the stress analysis to define the reliability of the system?
3. How do you provide the feedback of the moving parts. It's well-known that piezoelements have hysteresis effect. Also, do you check the linearity of teh actuators?
4. Moreover, could you specify the maximal load or blocking force of the presented system?
Please, comment these statements.
I think the description of the methods and technical aspects are presented properly. Meanwhile, English quality of the paper is acceptable.
Reviewer 2 Report
Dear Authors,
The paper should be revised before it is published. My comments and questions are listed in the attachment.
Kind Regards

The quality of English is fine.
Reviewer 3 Report
In my opinion, the manuscript is suitable for publication in Micromachines journal but Authors must complete a major revision. Manuscript should be revised according to following comments:
1. Chapter “Introduction” must be improved:
a) the authors must in the last paragraph specifically indicate the areas where the actuator design proposed by the authors will improve the operation of stick-slip actuators from literature that are previously cited in lines 46-59.
2. Chapter “Experiment and discussion” must be improved:
a) the authors need to describe the research methodology in more detail:
- what is the resolution of the laser sensor,
- the measurement accuracy should be determined,
- the parameters of the voltage amplifier should be presented, especially its linearity.
b) the connection of the piezoelectric stack with actuator elements should be described in more detail,
c) there is no discussion in the chapter at all. The presented results should be related to the results known from the literature. In the discussion, attention should be paid to:
- the waveforms and values of input voltages to piezoelectric stack,
- a description of the load acting on the actuator,
- the degree of complexity of the structure,
- method of connecting piezoelectric stacks with mechanical parts of actuators.
3. Chapter “Conclusions” must be improved:
a) authors must describe in which areas their prototype improves the state of the art. The authors proved that their prototype worked, but it is not known what this article contributed in a scientific sense.
Moderate editing of English language is needed.
Round 2
Reviewer 1 Report
Could be accepted in present form, but I recommend as before point various adaptive optics references in introduction and literature part.
Reviewer 2 Report
Dear Authors,
The manuscript can be published after minor revision. I do not have any other comments.
Kind Regards
English should be polished.
Reviewer 3 Report
I accept in present form.
Minor editing of English language required.